# Early Growth Response Gene Upregulation in Epstein–Barr Virus (EBV)-Associated Myalgic Encephalomyelitis/Chronic Fatigue Syndrome (ME/CFS)

**DOI:** 10.3390/biom10111484

**Published:** 2020-10-26

**Authors:** Jonathan Kerr

**Affiliations:** Department of Microbiology, Norfolk & Norwich University Hospital (NNUH), Colney Lane, Norwich, Norfolk NR4 7UY, UK; jonathan.kerr@nnuh.nhs.uk

**Keywords:** myalgic encephalomyelitis/chronic fatigue syndrome (ME/CFS), Epstein–Barr virus (EBV), EBV-induced gene 2 (EBI2), early growth response, EGR1, EGR2, EGR3, immediate early gene

## Abstract

Myalgic encephalomyelitis/chronic fatigue syndrome (ME/CFS) is a chronic multisystem disease exhibiting a variety of symptoms and affecting multiple systems. Psychological stress and virus infection are important. Virus infection may trigger the onset, and psychological stress may reactivate latent viruses, for example, Epstein–Barr virus (EBV). It has recently been reported that EBV induced gene 2 (EBI2) was upregulated in blood in a subset of ME/CFS patients. The purpose of this study was to determine whether the pattern of expression of early growth response (EGR) genes, important in EBV infection and which have also been found to be upregulated in blood of ME/CFS patients, paralleled that of EBI2. EGR gene upregulation was found to be closely associated with that of EBI2 in ME/CFS, providing further evidence in support of ongoing EBV reactivation in a subset of ME/CFS patients. EGR1, EGR2, and EGR3 are part of the cellular immediate early gene response and are important in EBV transcription, reactivation, and B lymphocyte transformation. EGR1 is a regulator of immune function, and is important in vascular homeostasis, psychological stress, connective tissue disease, mitochondrial function, all of which are relevant to ME/CFS. EGR2 and EGR3 are negative regulators of T lymphocytes and are important in systemic autoimmunity.

## 1. Introduction

Myalgic encephalomyelitis/chronic fatigue syndrome (ME/CFS) is a chronic multisystem disease characterized by at least six months of fatigue and various other symptoms, including headache, sore throat, muscle pain, joint pain, muscle weakness, post-exertional malaise, sleep abnormalities, with secondary anxiety and depression [1]. Alternative diagnostic criteria that better take account of the widespread inflammation and multisystemic neuropathology have also been developed [2]. Heterogeneity in ME/CFS is a well-established feature, although the precise nature of the subtypes and the means to identify them has been lacking. There are currently no biomarkers of ME/CFS per se. Thus, ME/CFS is a heterogeneous autoimmune-like disease with a variety of subtypes, similar to various other autoimmune diseases. Psychological (emotional) stress can trigger the disease, and anxiety and depression are secondary phenomena [1,2].

Psychological stress is common in ME/CFS, as it is in other autoimmune diseases, for example, ulcerative colitis, rheumatoid arthritis, psoriasis, asthma, etc. Such emotional aspects are believed to also underlie relapses and flare-ups which may precipitate hospital admission and the use of immune-modifying treatments in a variety of autoimmune diseases.

I previously reported that Epstein–Barr virus (EBV) induced gene 2 (EBI2) was upregulated in peripheral blood mononuclear cells (PBMC) of 12 of 31 (38%) patients with ME/CFS and that these patients appeared to have a more severe disease phenotype and lower levels of EBV nuclear antigen 1 (EBNA1) IgG [3]. EBI2 is a human G-protein coupled receptor (GPCR) which is powerfully induced by EBV infection [4], is upregulated in a variety of autoimmune diseases, and is important in immunity and central nervous system function [3].

The immediate early genes (IEG) and early growth responses 1–3 (EGR1–3) have been found to be upregulated in blood of a subset of patients with ME/CFS [5], in most of which EBI2 was also found to be upregulated [3], suggesting that these two phenomena are linked in ME/CFS. The purpose of this paper is to study the pattern of expression of the EGR genes in relation to that of EBI2 in ME/CFS and to determine whether they are likely to be involved in EBV reactivation in these patients. EGR genes were found to be co-expressed with EBI2 in ME/CFS. As EBI2 is a marker of EBV reactivation and as EGR1–3 are important for EBV reactivation in B lymphocytes (see below), these findings provide further support for the existence of an EBV-induced subtype of ME/CFS, and highlight, for the first time, the importance of a chronic immediate early gene response in EBV-associated ME/CFS.

## 2. Early Growth Response (EGR) Gene Upregulation in Myalgic Encephalomyelitis/Chronic Fatigue Syndrome (ME/CFS)

We have previously reported that EGR1–3 and EBI2 were upregulated in PBMC of ME/CFS patients as compared with normal blood donors [5]. These genes were identified in ME/CFS through analysis of gene expression in PBMC using microarray and reverse transcriptase polymerase chain reaction (RT-PCR) confirmation assays which revealed differential expression of 82 human genes. Analysis of the promoter sequences of these 82 human genes revealed over-representation of binding sites for the following transcription factor genes: EGR1, EGR2, EGR3, EGR4, SP1, REPIN1, ETS1, GABPA, GTF3A, NFKB1, NHLH1, REST, and the Epstein–Barr virus (EBV) R transactivator gene BRLF1. Real-time PCR of these genes confirmed differential expression of EGR1, EGR2, EGR3, SP1, ETS1, REPIN1, GABPA, NHLH1, and NFKB1 (Table 1) [5]. This suggested that upregulation of these genes was key to the overall gene signature observed in the ME/CFS group. EGR1, EGR2, EGR3, and SP1 were found to be upregulated in a subset of ME/CFS patients, while the remaining transcription factors were upregulated in all or most of the ME/CFS patients (Table 1).

EBI2 mRNA was found to be upregulated in 12 of 31 (38%) ME/CFS patients and none of the controls [3]. Because EBI2 has been found to be highly expressed in PBMC (B, T, NK, monocytes, and granulocytes) during EBV reactivation [4,6,7], this marker has identified an EBV-induced subtype of ME/CFS [3].

Upregulation of EGR1, EGR2, and EGR3 occurred in ME/CFS patients with a raised level of EBI2 mRNA (Figure 1), suggesting that expression of these four genes in PBMC of ME/CFS patients was linked with EBV infection/reactivation (see below). Furthermore, the pattern of expression of EGR1, EGR2, EGR3, and EBI2 genes, in these patients, showed that patients with a high level of EBI2, also exhibited marked upregulation of EGR1, EGR2, and EGR3 (Figure 1), implying that the EGR gene upregulation was linked with EBI2 upregulation, and maybe related to EBV infection/reactivation in these patients. In those ME/CFS patients with marked upregulation of EBI2, the pattern of expression of EGR genes was similar, with levels of mRNA expression occurring in the order, from highest to lowest, EGR1, EBI2, EGR2, EGR3 (Figure 1).

## 3. Cellular Immediate Early Gene (IEG) Response

The cellular as distinct from viral IEG response, was first discovered in cultured cells as those genes that were induced in response to growth factors and mitogenic agents, and were later shown to occur in most cell types in response to a variety of stimuli. The cellular IEG response is a rapid and transient increase in mRNA encoding the IEGs, independent of protein synthesis, and requiring only post-translational modification of pre-existing molecules. It is assumed that the cellular IEG response regulates downstream responses by controlling expression of early target genes. Cellular stimuli which trigger an IEG response include raised glucose, hypoxia, UV irradiation, mechanical stimulation, hormones (including insulin), growth factors, cytokines, and infection with a wide variety of viruses [8]. IEGs are critical mediators of gene–environment interactions and have been termed the “gateway to the genomic response” [8]. Forty cellular IEGs have been identified, including c-fos, c-myc, and c-jun, which are homologous to retroviral oncogenes, and are early regulators of cell growth and differentiation, as well as many other cellular processes. Many of the IEGs are transcription factors and DNA-binding proteins, while others are secreted proteins, cytoskeletal proteins, and receptor subunits.

This rapid cellular response to biological or environmental signals is critical for survival and adaptation of an organism [9,10,11]. Inducible IEG expression underlies acute inflammation [12,13,14,15], neuronal activity [16], cell proliferation, and differentiation [9,17,18]. Aberrant IEG expression is a feature of malignant cellular transformation [19,20,21]. IEG promoters exhibit enrichment of active chromatin marks and accumulation of RNA polymerase II in a poised fashion [22]. Chromatin remodelling at IEG promoters exposes specific transcription factor binding sequences such as nuclear factor kappa B (NF-KB), serum-response factor (SRF), and cyclic AMP response element-binding protein (CREBBP) [23].

## 4. Early Growth Response (EGR) Genes

The early growth response (EGR) family is comprised of four members (EGR1–4) which are C2H2-type zinc finger proteins and function as transcriptional regulators. EGRs1–4 are highly homologous within and between species, particularly within the region containing three C2H2 zinc finger DNA binding domains, suggesting possible overlap in targets and function of the EGR proteins. In contrast, the N-terminal regions of the EGR proteins differ significantly. EGR1–3 have a domain of interaction with transcriptional co-repressors, NAB1 and NAB2, suggesting that these proteins can mediate transcriptional repression, as well as negative regulation of the EGR proteins themselves [24].

Early growth response (EGR) genes have an extremely broad and diverse influence on many aspects of human physiology and disease. Significant themes which have been identified are the immune response and its relevance in systemic autoimmunity; vascular homeostasis; psychological stress and secondary physiological responses; stress-related mood disorders and schizophrenia; drug reward, withdrawal and relapse; connective tissue physiology and disease; and cancer biology. These aspects are summarized in Table 2.

### 4.1. EGR1

EGR1 is the best characterised among the EGR gene family. It is an 80kDa multifunctional transcription factor which upregulates genes involved in cell growth, cell cycle regulation, differentiation, mitogenesis, and tumor suppression. EGR1 is a marker of neuronal activity which allows mapping of brain activation following a specific behavioural, pharmacological or environmental event [66]. Induction of p38 and ERK/MAPK pathways lead to activation of ELK1 and CREB transcription factors which, then, bind their respective response elements in the EGR1 promoter. Binding sites for several key transcription factors are also found on the EGR1 promoter, including Sp1, AP1, NFKB and EGR1 itself. ELK1, CREB, and EGR1 interact to rapidly and transiently activate EGR1 transcription. However, CREB, ELK1, SRF, and RNA polymerase (Pol II) are present at the promoter prior to its induction [67], suggesting that similar to other IEGs [23], EGR1 transcription is poised at baseline. EGR1 promoter-bound CREB and ELK1 are phosphorylated in a p38 and MEK1/2-dependent manner [68]. This results in increased phosphoacetylation and acetylation of histone H3 at the +1 nucleosome [67], which is likely mediated by the histone acetyltransferase (HAT) activity of CREB-binding protein (CREBBP) transcriptional cofactor, as its binding to the mouse EGR1 promoter increases in parallel with its transcription [68]. This results in increased RNA Pol II recruitment, promoting rapid EGR1 transcription. EGR1 regulation by histone acetylation and methylation has also been demonstrated in neurons in vivo as events underlying learning, memory, cognition, and response to stress [69,70,71]. EGR1 regulation by DNA methylation and hydroxymethylation also occurs with aging in the rat hippocampus [72] and with sleep deprivation in mouse cortex [73]. Therefore, post-translational modifications are critical regulators of EGR1 activity and stability.

There are well documented sex differences in expression of EGR1 in different parts of the rat brain [66,74,75]. This may be due to the fact that oestrogen may directly upregulate EGR1 expression in different organs [76]. Although there is an oestrogen response element in the EGR1 promoter, oestrogen-induced EGR1 transcription is mediated by SRF and ELK1 binding to the SRE site [77,78].

Regarding EGR1 gene targets, the nine-nucleotide long sequence GCGG/TGGGCG was originally identified as the EGR1 recognition sequence [79,80]. Subsequent analysis of the EGR1 binding sequence revealed variation in this sequence and revealed an optimal site of at least 10 nucleotides [81]. EGR1 can also regulate gene expression through interaction with other transcription factors including c/EBPb, Fos, and Jun [36,82,83,84], which hugely expands its influence. EGR1 has also been shown to directly regulate 124 distinct miRNAs and 63 pre-miRNAs in human erythroleukaemia cells (K562) [85]. Among these, miR-124, is a known regulator of EGR1 [85,86].

The Encyclopedia of DNA Elements (ENCODE) project included EGR1 as part of the tier 1 chromatin immunoprecipitation followed by deep sequencing (ChIP-seq), which provided vast information on EGR1 target genes [87]. Among 1.5872 × 10^4^ annotated genes, 8552 (53.9%) contained at least one EGR1 binding region within 3 kb of their transcription start site (TSS), indicating that in diverse human cells, EGR1 binds a huge number of genes and potentially regulates a very large gene expression profile. Functional analysis of genes containing at least one EGR1 binding site, has revealed enrichment of pathways related to growth factor signalling and general intracellular signalling such as Ras and MAPK, also regulating EGR1 expression itself. Molecular functions include chromatin binding, transcription factors, guanyl-nucleotide exchange factor activity, and serine/threonine kinase activity, revealing that EGR1 controls every level of signal transduction, from second messenger to transcription factor. Cellular localization of the EGR1-bound gene products ranges from chromatin to the cell membrane. These data suggest that EGR1 regulates cell-cell communication through regulation of a vast array of genes [66].

### 4.2. EGR2 and EGR3

EGR2 expression is associated with the onset of myelination in the peripheral nervous system and hindbrain development [88], and EGR3 plays a critical role in muscle-spindle development [89]. EGR2 and EGR3 are indispensable for T and B cell development and activation. EGR2 and EGR3 are expressed during selected stages of B and T cell development in the thymus and bone marrow, respectively [90]. Their expression is markedly reduced in immature single positive (ISP) thymocytes and pro-B cells, suggesting that they are indispensable for development of double negative (DN) cells following β selection. Double negative lymphocytes are those which carry α and β chains of the T cell receptor, but which lack CD4 and CD8 markers [91]. EGR2 expression is halted after β chain selection for immature single positive (ISP) cell differentiation to form double positive (DP) cells [90]. Double positive cells express both CD4 and CD8 markers during T cell development [91]. T and B cells with EGR2 and 3 deleted show severe defects in proliferation and IL-2 production following antigenic receptor stimulation [92].

EGR2 is a cell growth suppressor. EGR2 is likely to be a mediator of PTEN growth-suppressive signalling [93], in which cytoplasmic PTEN, acts as a growth suppressor, induces EGR2 expression which, then, transactivates BNIP3L and BAK (pro-apoptotic proteins of the Bcl-2 family), which alter the permeability of mitochondrial membranes, to release cytochrome c and activate caspases 3, 8, and 9 [94]. Therefore, it has been thought that exogenous EGR2 gene therapy could be used to treat diverse cancers. EGR2 has been found to be growth inhibitory when overexpressed in neoplastic cells through induction of CEBPB transcription factor which contributes to oncogene-induced senescence in fibroblasts expression oncogenic Ras or BRAF from different species [95].

EGR2 also suppresses the c-Jun NH2-terminal protein kinase (JNK)-c-Jun pathway, which is required for proliferation and death [96]. Therefore, EGR2 can coordinately control the suppression of cell division and cell death via the inactivation of the JNK-c-Jun pathway. In contrast, EGR2 knockdown resulted in inhibition of cellular proliferation and spheroidal growth in vitro, and induced regression of Ewing sarcoma xenografts, reduced progression through the S phase of the cell cycle, and reduced cell viability [97].

## 5. Early Growth Response (EGR) Genes in Immunity and Systemic Autoimmunity

The immune mechanisms for maintenance of tolerance to self-antigens and prevention of autoimmunity include anergy, apoptosis, and function of regulatory T cells [98,99]. Both EGR2 and EGR3 can induce anergy both in vitro and in vivo [100,101,102]. EGR2 and EGR3 negatively regulate T cell activation and block upregulation of NAB2 and EGR1 which inhibits T cell function [102]. EGR1 with NAB2 as coactivator, stimulates IL-2 production [103]. In cells overexpressing EGR2 and EGR3, Cbl-b was increased, and T cells from EGR3-/- animals had much reduced Cbl-b as compared with wild-type mouse controls. This supports the view that EGR2 and EGR3 can upregulate genes that inhibit T cell activation [104]. Silencing EGR2 expression prevents full development of anergy and increases responsiveness of CD3 + CD28- cells to stimulation, which results in restoration of proliferation and IL-2 production [105]. EGR2 induces Ndrg1 which is a T cell clonal anergy factor, which is upregulated by anergic signalling, and is maintained at a high level, in resting anergic T cells, which mediates reduction in T cell mediated inflammation [106].

Null mutation of each of EGR2 and EGR3 results in death in 100% and approximately 40% of mice, respectively, during the perinatal period [107,108], which makes it difficult to study the influence of EGR2 and EGR3 on the immune system. Therefore, EGR2 was deleted in T and B lymphocytes only in so-called EGR2 conditional knockout (CKO) mice, and it was shown that these mice developed lupus-like syndromes during maturity [109]. They exhibited hair loss, skin lesions, and massive multi-organ infiltration with mononuclear cells, along with high titre anti-dsDNA and anti-histone autoantibodies. EGR2 transcripts were highly expressed in CD4 + CD44high T cells, and EGR2 directly regulated the expression of cyclin-dependent kinase inhibitor p21cip1 to suppress the proliferation of CD4 + CD44high T cells. EGR2-deficient CD4+ T cells produced greater amounts of IFN-γ and IL-17 in response to stimulation of the TCR. Expression of T-bet (Th1 transcription factor), IL-6, IL-21, and IL-23 (inducers of Th17 differentiation), were not altered in CD4+ T cells of mature Egr2 CKO mice. The study of EGR2 CKO mice suggests that EGR2 limits systemic autoimmunity through suppression of effector T cell proliferation and cytokine production.

Double knockout (DKO) mice, which were deficient in both EGR2 and EGR3 in B and T lymphocytes, developed early onset, lethal systemic autoimmune syndrome with multi-organ lymphocyte infiltration, high titre autoantibody to dsDNA and histones, along with severe glomerulonephritis [92]. These mice also had high levels of serum IL-6, IL17A, Il-17F, and IFN-γ, along with expanded populations of IL-17 and IFN-γ producing CD4+ cells. Proliferation of B and T cells was markedly impaired upon antigen challenge in vitro. In addition, B and T cells from EGR2/3 DKO mice showed STAT1 and STAT3 activation with reduced expression of suppressor of cytokine signalling 1 (SOCS1) and 3 (SOCS3). SOCS1 suppresses STAT1 activation and inhibits Th1 differentiation. Mice deficient in T cell SOCS1 did not develop spontaneous inflammatory disease despite their T cells differentiating into Th1 cells which produced high level IFN-γ through STAT1 overexpression [110]. SOCS3 suppresses STAT3 activation which negatively regulates Th17 cell differentiation. Mice lacking SOCS3 in lymphocytes developed late-onset systemic inflammatory disease, while their T cells were shown to be more susceptible to IL-6- and IL-23-induced differentiation of Th17 cells [111]. EGR2 directly binds the SOCS1 and SOCS3 promoters inducing expression of SOCS1 and SOCS3 which negatively regulate STAT1- and STAT3-mediated Th1/Th17 polarization. B and T cells from EGR2/3 DKO mice also revealed decreased IL-2 production [92] and were shown to be severely deficient in activation of conventional AP-1, which was essential for IL-2 production and cell cycle progression. Various findings have indicated that EGR2 and EGR3 support AP-1 activation through suppression of basic leucine zipper transcription factor ATF-like (BATF), a member of the AP-1 family of transcription factors [112,113].

In EGR2 CKO mice, EGR3 may compensate for the lack of EGR2 function as the phenotype of these mice is milder than that of EGR2/3 DKO mice. EGR3 CKO mice, however, do not develop autoimmune symptoms [92], likely due to compensation of EGR2 for deficient EGR3. There are similarities of function of EGR2 and EGR3. For example, both induce IL-10 expression in mouse CD4+ T cells [114]. EGR3 induces TGF-β1 expression in CD4+ T cells and increases phosphorylation of STAT3, a feature that is associated with transcription of TGF-β1; CD4+ T cells which are deficient in EGR3 produce lower levels of TGF-β1. EGR3 regulates mouse CD4+ T cells in an antigen-specific manner. EGR3-transduced CD4+ T cells from mice with collagen-induced arthritis and delayed-type hypersensitivity were shown to have powerful regulatory activity in vivo. CD4+, CD25, CD45RO, and LAG3 T cells inhibited graft-versus-host disease in immunosuppressed mice with transplanted human peripheral blood mononuclear cells (PBMC) [115]. Therefore, EGR3 is a transcription factor linked with TGF-β1 expression and regulatory activity in vivo in both humans and mice. TGF-β1 induces apoptosis of B lymphocytes with suppression of immunoglobulin production [116]. Mice lacking TGF-β1 exhibit severe autoimmunity with high titre antibody to nuclear antigens, and deposition of immune complexes in the glomerulus [117].

Low circulating TGF-β1 is a consistent finding in humans with systemic lupus erythematosus (SLE) and is correlated with disease activity [118]. PBMC from SLE patients produce significantly less TGF-β1 than PBMC from normal controls [119]. Therefore, EGR3-mediated TGF-β1 upregulation in T lymphocytes has been suggested for treatment of autoimmune disease in general [120].

The NZM2410 mouse strain is a cross between the lupus-prone New Zealand white (NZW) (lupus prone) and New Zealand black (NZB) lineages. NZM2410 mice develop early progression of anti-nuclear antibody (ANA) severe lupus gomerulonephritis at an early age. In NZM2410 mice, the Sle1b locus mediates ANA production and the Ly108 gene of the signalling lymphocytic activation molecule (SLAM) family is the strongest candidate gene at this locus [121]. The Ly108.1 allele is highly expressed in early immature B lymphocytes from lupus prone B6.Sle1b mice. The normal Ly108.2 allele sensitizes immature B lymphocytes to deletion. In addition, EGR2 polymorphisms influence SLE susceptibility in humans [122], suggesting that Ly108 regulates autoimmunity through EGR2 induction.

## 6. Early Growth Response (EGR) Genes in Epstein–Barr Virus (EBV) Infection

The EGR genes 1–3 are critically important in EBV infection. Table 3 summarises key aspects of the role of EGR genes in EBV infection in terms of EBV transcription, lytic cycle, and B lymphocyte transformation.

### 6.1. EBV Transcription

A GC-rich sequence in the −99/−91 EBV C promoter (Cp) region was identified as essential for oriPI-EBNA1-independent, as well as oriPI-EBNA1-dependent, activity of the promoter. This region contains overlapping binding sites for SP1 and EGR1. Results suggest that SP1 is a positive regulator and EGR1 is a negative regulator of Cp activity in B lymphoid cell lines. Bound EGR1 prevents binding of the positive regulator, SP1 [123].

### 6.2. Lytic Cycle

Following the acute phase, Epstein–Barr virus (EBV) establishes latency in memory B lymphocytes [133] and other lineages [134,135]. EBV reactivation from latency can occur in both memory B lymphocytes and cancer cells, in response to a variety of physiological stimuli [136]. Zta (BZLF1, ZEBRA) is the key regulator of EBV lytic replication and is an EBV-encoded transcription/replication factor which interacts directly with DNA via its basic region in both Zta-regulated promoters and within the origin of lytic replication [137,138,139,140,141,142]. Zta also interacts indirectly with DNA and influences gene expression through interaction with various host transcription factors [143,144,145,146,147,148].

Host EGR1 transcription is induced following EBV lytic cycle activation, through the action of Zta [125]. Activation of EGR1 transcription by Zta occurs through a serum response element (SRE) that is flanked by two Ets response elements and a pair of potential Zta response elements (ZREs). Zta activates Erk of the MAPK family with transactivation occurring through the Ets response elements [125]. EGR1 is methylated in B cells, but treatment with a combination of demethylation agents and physiological stimuli can induce EGR1 expression [149], thus, demonstrating that methylation plays a role the control of EGR1 gene expression.

Expression of EGR1, as well as EGR2 and EGR3, has been shown to occur upstream of EBV BRLF1 and BZLF1 following cross-linking of the B lymphocyte receptor in Burkitt lymphoma cells [128].

### 6.3. B lymphocyte Transformation

EGR1 expression has been shown to correlate with the transformed phenotype and type of viral latency in EBV+ lymphoid cell lines [129]. EGR1 expression is increased by EBV LMP1 via NFKB and is required for EBV latent membrane protein 1 (LMP1)-induced cancer cell survival [130]. LMP1 is the primary transforming product of EBV. LMP1 is expressed in most EBV-associated lymphoproliferative diseases and malignancies. LMP1 subverts cellular signal transduction to achieve cell transformation and immortalisation, but is also responsible for cytokine induction, immune modulation, regulation of tumour angiogenesis, cell-cell contact, cell migration, and invasive growth of tumour cells [150]. In B cell lymphoma, EBV LMP1 increases genomic instability through EGR1-mediated upregulation of activation-induced cytidine deaminase [132]. In EBV+ Hodgkin lymphoma, EBER1 inhibits p21cip1/waf1 transcription and prevents apoptosis through downregulation of p53, EGR1, and STAT1 [131].

## 7. Relevance of EGR-Associated Pathology to ME/CFS

There is a significant overlap among the pathological processes affected by EGR gene abnormalities and those associated with ME/CFS (summarised in Table 4).

Virus infections are well-recognised triggers of ME/CFS [151] and are also known to lead to rapid upregulation of EGR1 as part of an immediate early gene response. More specifically, EBV is an important trigger of ME/CFS [3], which upregulates EGR1, EGR2, and EGR3 in B lymphocytes [125,128].

The EGR genes are critically important for the immune response; EGR1 is an important regulator in lymphocytes [25], while EGR2 and EGR3 are negative regulators involved in anergy induction and apoptosis [28,120]. In ME/CFS, NK cells are consistently implicated with significant increases in iNKT cell phenotypes reported [153]. In addition, raised circulating TGF-β1 has been implicated by five of eight studies [154], despite which ß2 adrenergic (ß2AdR) and M3 acetylcholine receptor autoantibodies have been found to be elevated in a subset of ME/CFS patients [155,156].

EGR1 is a critical mediator of revascularisation after vascular occlusion [29,30]. ME/CFS patients exhibit numerous, well-documented vascular abnormalities, including hypovolaemia, venous pooling, reduced sodium reabsorption, orthostatic dysfunction and postural tachycardia syndrome (POTS), and dominant vagal tone [157].

In animals, acute psychological stress (restraint, immobilization or forced swim) leads to increased EGR1 mRNA in multiple brain locations, and EGR1 is critical in encoding the chronic behavioural effects of stress [66]. In ME/CFS, psychological stress is a prominent feature, which may facilitate virus acquisition, development of symptomatic as opposed to asymptomatic infection, and in reactivation of EBV [158,159].

EGR1 regulates matrix production in various connective tissues and is often involved in abnormal production of extracellular matrix in fibrotic conditions [8]. Connective tissues are prominently affected in ME/CFS, although this may be due to systemic effects [1,2].

EGR1 binding sites occur in promoters of several important mitochondrial genes and increases upon skeletal muscle contraction. In addition, changes in intracellular calcium modify mitochondrial phenotype, in part via the involvement of EGR1 [41,42,160]. ME/CFS patients exhibit impaired cellular metabolism [160,161]. ME/CFS T cells exhibit hypometabolism [162].

Finally, stress-related mood disorders, schizophrenia, drug-related aspects, and cancer, in which EGR1 plays important roles, have no counterparts in ME/CFS.

## 8. Conclusions

In conclusion, ME/CFS is a multisystem disease which can be triggered by infection with any of a range of viruses, including EBV and other herpesviruses such as HHV6, parvovirus B19, and others [151]. The present report documents the upregulation of EGR1, EGR2, and EGR3 in ME/CFS patients who exhibit upregulation of EBI2 gene. As these patients have had ME/CFS for at least six months, and in some cases, several years, these data support an ongoing process driven by chronic EBV reactivation. Important aspects for further study would be the determination of EBI2 and EGR gene expression in the different subsets of PBMC in ME/CFS patients and controls, the study of these genes over time in individual ME/CFS patients, and the study of phenotypic differences in ME/CFS patients with and without upregulation of EBI2 and EGR genes, to confirm our previous study which suggested that the EBI2 subtype was more severely affected than other non-EBI2 ME/CFS patients.

I have previously proposed that upregulation of the EBI2 gene could be used to identify an Epstein–Barr virus induced subtype of ME/CFS which may have increased severity. Although the EGR genes are upregulated in some ME/CFS patients, the issue of whether they can be used as biomarkers is unclear. However, I do not believe that the EGR genes would be useful as biomarkers for either ME/CFS (as they are upregulated only in a subset) or EBV-associated ME/CFS (as EBI2 is likely to be more specific).

## Figures and Tables

**Figure 1 biomolecules-10-01484-f001:**
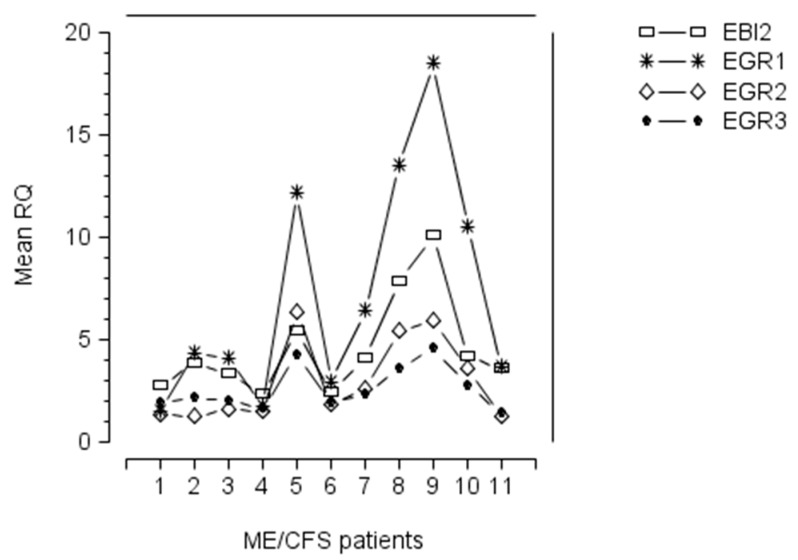
Expression of Epstein–Barr virus (EBV) induced gene 2 (EBI2) (open rectangle), early growth response gene 1 (EGR1) (cross symbol), early growth response gene 2 (EGR2) (open diamond), and early growth response gene 3 (EGR3) (solid circle) mRNAs in peripheral blood mononuclear cells (PBMC) of 11 myalgic encephalomyelitis/chronic fatigue syndrome (ME/CFS) patients with previously documented upregulation of EBI2 gene [3]. The figure shows that ME/CFS patients with a high level of EBI2, also exhibit upregulation of EGR1, EGR2, and EGR3, implying that the EGR gene upregulation is linked with that of EBI2. In those ME/CFS patients with marked upregulation of EBI2, there was similarly marked upregulation of the EGR genes, with level of mRNA expression occurring in the order, from highest to lowest, EGR1, EBI2, EGR2, EGR3.

**Table 1 biomolecules-10-01484-t001:** Transcription factors found to be upregulated in peripheral blood mononuclear cells (PBMC) of patients with Myalgic encephalomyelitis/chronic fatigue syndrome (ME/CFS) as compared with normal blood donors by real-time reverse transcriptase polymerase chain reaction (RT-PCR) [5].

Gene Symbol	Gene Name	GenBank Accession No.	Real-Time PCR Fold-Difference	2-Tailed P (RT-PCR)
**EGR1**	Early growth response 1	NM_001955	2.82	0.015 ^†^
**EGR2**	Early growth response 2	NM_000399	2.32	0.09 ^†^
**EGR3**	Early growth response 3	NM_004421	1.92	0.017 ^†^
**SP1**	Sp1 transcription factor	NM_138473	2.47	0.11 ^†^
**ETS1**	ETS proto-oncogene 1, transcription factor	NM_005238	2.11	1 × 10^−5 ‡^
**REPIN1**	Replication initiator 1	NM_013400	3.62	6 × 10^−6 ‡^
**GABPA**	GA binding protein transcription factor Subunit alpha	NM_002031	8.06	3 × 10^−4 ‡^
**NHLH1**	Nescient helix loop helix 1	NM_005589	11.51	7 × 10^−4 ‡^
**NFKB1**	Nuclear factor kappa B subunit 1	NM_003998	1.59	4.04 × 10^−5 ‡^

^†^ Genes which were upregulated in a subset of ME/CFS patients. ^‡^ Genes which were upregulated in most ME/CFS patients.

**Table 2 biomolecules-10-01484-t002:** Early growth response (EGR) genes in physiological and pathological responses.

Function of EGR Genes in Physiological and Pathological Responses	References
**Virus Infection**	
EGR1 is a key mediator in the response to and pathogenesis of many different viruses that infect humans, including those of the herpesviruses, retroviruses, flaviviruses, poxviruses, hepatitis B virus, and Borna disease virus; for Epstein–Barr virus, see Table 3	See numerous papers in PubMed
**Immune Response and Inflammation**	
EGR1 is an important regulator of the immune response and the differentiation pathway of myeloid precursors	[25]
Promitotic genes in B lymphocytes include EGR1 and EGR2	[26]
EGR1 functions as a positive regulatory factor in B and T cells via transcription of key cytokines and costimulatory molecules EGR2 and -3 act as negative regulators involved in anergy induction and apoptosis	[27,28]
**Vascular Homeostasis**	
EGR1 plays a pivotal role in reperfusion responses to vascular occlusion in mice and possibly other mammals	[29]
EGR1 is a critical and potentially therapeutic mediator of revascularization after vascular occlusion and implicates arteriogenesis (collateral vessel formation) as a critical component of this process	[30]
**Psychological Stress**	
In animal experiments, acute stress (restraint, immobilization, or forced swim), leads to increased Egr1 mRNA in multiple locations throughout the brain; these include neocortical areas, hippocampus, lateral septum, caudate putamen, nucleus accumbens, amygdala, and paraventricular nucleus (PVN) of the hypothalamus	[31,32,33,34,35,36,37]
EGR1 is critical in encoding the chronic behavioural effects of stress, for example, acute exposure to forced swim stress or activation of the glucocorticoid receptor (GR) upregulates EGR1 in the rat and mouse hippocampus, mediating stress-related fear memories	[38,39,40]
**Connective Tissue Disease**	
EGR1 is expressed in tendon, cartilage, bone and adipose tissue, and is involved in development, homeostasis, and healing process of these tissues, via regulation of the extracellular matrix EGR1 is often involved in the abnormal production of extracellular matrix in fibrotic conditions such as systemic sclerosis, and EGR1 deletion may be therapeutic for these conditions	8
**Mitochondrial Function**	
Egr1, Sp1, and SRF are potentially important in mitochondrial biogenesis as their binding sites occur in promoters of important mitochondrial genes; mouse skeletal muscle contraction is associated with marked increase in Egr1 mRNA expression; changes in intracellular Ca(2+) can modify mitochondrial phenotype, in part via the involvement of Egr1	[41,42,43]
**Stress-Related Mood Disorders and Schizophrenia**	
In prefrontal cortex of cadavres of patients with major depression as compared with normal controls, EGR1 is lowered; low EGR1 in PFC was observed in those untreated and also treated but not responding to treatment	[44]
In schizophrenia patients, EGR1 mRNA is downregulated in dorsolateral PFC compared with controls	[45,46]
EGR1 mRNA in the PFC of schizophrenia patients is positively correlated with glutamic acid decarboxylase 1 (GAD1) mRNA, which is a robust molecular feature of schizophrenia	[46,47]
Reduced EGR1 expression is observed in the PVN, mPFC, HPC, or extended amygdala of rats, mice, and prairie voles following social isolation	[48,49,50,51,52,53,54]
**Drug Reward, Withdrawal and Relapse**	
Injection of both heroin and morphine upregulates Egr1 mRNA in the nucleus accumbens, dorsal striatum, and cingulate cortex of C57Bl6 mice	[55,56,57,58]
In rats and mice, acute ethanol exposure increases EGR1 expression in multiple brain areas including the PFC, amygdala, supraoptic nucleus, hippocampus and nucleus accumbens	[59,60,61,62,63]
**Cancer**	
EGR1 regulates angiogenic and osteoclastogenic factors in prostate cancer and promotes metastasis	[64]
EGR1 is aberrantly expressed in various cancers, regulating tumour cell proliferation, apoptosis, migration, invasion, and tumour microenvironment.	[65]

**Table 3 biomolecules-10-01484-t003:** Early growth response (EGR) genes in Epstein–Barr virus (EBV) infection.

Description	Reference
**EBV Transcription**	
EGR1 is a negative regulator of the EBV C promoter in B lymphoid cell lines EGR1 prevents binding of the positive regulator, SP1	[123]
**EBV Lytic Cycle**	
EGR1 activates expression of BRLF1 via binding to the BRLF1 promoter	[124]
EGR1 expression is induced by EBV transactivator Zta	[125]
EBV transactivator Zta interacts with methylated ZRE in the EGR1 promoter	[126]
Suppression of LMP2A target gene, EGR1, protects Hodgkin’s lymphoma cells from entry into lytic cycle	[127]
Cellular immediate early gene (EGR1, EGR2, and EGR3) expression occurs kinetically upstream of EBV BRLF1 and BZLF1 following cross-linking of the B cell receptor in Burkitt lymphoma cells	[128]
**EBV-Induced B Lymphocyte Transformation**	
EGR1 expression correlates with the transformed phenotype and type of viral latency in EBV+ lymphoid cell lines	[129]
Expression of EGR1 was increased by EBV LMP1 via NFKB, and is required for LMP1-induced cancer cell survival	[130]
In EBV+ Hodgkin lymphoma, EBER1 inhibits p21cip1/waf1 transcription and prevents apoptosis through downregulation of p53, EGR1, and STAT1	[131]
In B cell lymphoma, EBV LMP1 increases genomic instability through EGR1-mediated upregulation of activation induced cytidine deaminase	[132]

**Table 4 biomolecules-10-01484-t004:** Relevance of EGR-Associated Pathology to Myalgic Encephalomyelitis/Chronic Fatigue Syndrome (ME/CFS).

EGR-Associated Pathology	Relevance to ME/CFS	References
**Virus Infection**	Numerous viruses have been reported to trigger ME/CFS	[151]
	EBI2 gene upregulation was reported in approximately 50% of ME/CFS patients in one study; EBI2 upregulated patients were severely affected; EBI2 upregulation indicates EBV reactivation	[3]
**Immune Response and Inflammation**	Increased effector memory CD8+ T cells and decreased terminally differentiated effector CD8+ T cellsSignificantly increased proportion of mucosal associated invariant T cells (MAIT) cells, especially in severely affected ME/CFS patients	[152]
	Severe CFS/ME patients differed from controls and moderate CFS/ME patients over time and expressed significant increases in iNKT cell phenotypes, naive CD8+T cells, and γδT cells with significant reduction in NKG2D receptors at 6 months.	[153]
	Raised circulating transforming growth factor-β1 has been found in five of eight studies	[154]
	ß2 Adrenergic (ß2AdR) and M3 acetylcholine receptor autoantibodies have been found to be elevated in a subset of ME/CFS patients; removal of these autoantibodies by IgG apheresis led to rapid improvement in most patients demonstrating a pathophysiological role of autoantibodies in ME/CFS	[155,156]
**Vascular Homeostasis**	ME/CFS patients exhibit numerous, well-documented vascular abnormalities, including hypovolaemia, venous pooling, reduced sodium reabsorption, orthostatic dysfunction and postural tachycardia syndrome (POTS), and dominant vagal tone; these abnormalities are likely caused by ß2 adrenergic (ß2AdR) and M3 acetylcholine receptor autoantibodies	[157]
**Psychological Stress**	Psychological stress has been shown to be important in virus transmission, development of symptoms following virus acquisition, as a predisposing factor during the onset of CFS/ME, and in reactivation of EBV	[158,159]
**Connective Tissue Disease**	Pain in muscles and joints are a well-recognised feature of ME/CFS, however, this is likely to be due to systemic effects	[1,2]
**Mitochondrial Function**	Impaired cell metabolism has been documented in ME/CFS	[160,161]
	ME/CFS CD8+ T cells exhibit reduced mitochondrial membrane potential as compared with those from healthy controls; both CD4+ and CD8+ T cells from patients with ME/CFS exhibit reduced glycolysis at rest, whereas CD8+ T cells also exhibit reduced glycolysis following activation; in addition, proinflammatory cytokines correlated with hypometabolism in T cells	[162]
**Stress-Related Mood Disorders and Schizophrenia**	None	
**Drug Reward, Withdrawal and Relapse**	None	
**Cancer**	None

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
