# Peer review of "Early Growth Response Gene Upregulation in Epstein–Barr Virus (EBV)-Associated Myalgic Encephalomyelitis/Chronic Fatigue Syndrome (ME/CFS)"

_biomolecules, 2020, doi:10.3390/biom10111484_

Round 1

Reviewer 1 Report

This is a well written review concerning the upregulation of the host early response genes (Egr1, Egr2, and Egr3) in a cohort of patients with Myalgic Encephalomyelitis/Chronic Fatigue Syndrome (ME/CFS). The author utilizing data obtained in his previous studies suggests that in a cohort of patients with ME/CFS there is a significant elevation of Epstein-Barr virus (EBV) induced gene 2 (EBI2) and that this correlates with an increased upregulation of the expression in the transcription factors Egr1, Egr2 and Egr3 in this patient population. Interestingly, this patient population also appears to exhibit a more severe disease phenotype. The remainder of the manuscript is a review of the roles of the early response genes in various cellular processes, including the life cycle of EBV and in their possible role in contributing to the symptomology observed in ME/CFS patients. In my opinion the manuscript could be enhanced by including information concerning any evidence of how these early response genes contribute to the replication of EBV and by expanding the Conclusion section to address whether or not the data supports the use of the expression of these growth response genes as potential biomarkers in this cohort of patients.

Author Response

Thankyou for your review. 

Information on the influence of EGR genes on EBV replication has been included in Table 3, under 'EBV lytic cycle', and summarised on page 11 under section 6 (EGR genes in EBV infection). 

In response to your suggestion, I have added a paragraph to the conclusion to address the issue of whether the EGR genes could be used as biomarkers of ME/CFS. I don't think they would be so useful as biomarkers, but they seem to provide support for an ongoing role for EBV in the pathogenesis of some cases of ME/CFS. As ME/CFS is heterogeneous, I believe this is very interesting. 

Reviewer 2 Report

It is a well-written review.  Concerning the upregulations, the host early genes in a cohort of CFS/ME patients, it would be my question that since EBV is acquired early in life, and then the virus becomes dormant, how did you select the CFS/ME patients with EBV that you studied for the above responses? It would be interesting to mention this in the discussion section. Did you select the EBV patient by PCR or by an immunity test for antibodies such as EA or VCA?

The manuscript could be enhanced by information concerning any evidence of how these early response genes contribute to the replication of EBV and by expanding conclusions and actions to address whether or not the data supports use of the expression of the growth response genes as potential biomarkers in this cohort of patients.

For many years, other viruses besides EBV have been listed as contributors to CFS/ME pathogenesis. There is good evidence that besides EBV, human herpesvirus 6 may trigger the CFS patient's responses that may correlate with the symptomology.  As a suggestion, I thought you might consider reading this paper with this information, and reference it in your review (Rasa, Santa et al. “Chronic viral infections in myalgic encephalomyelitis/chronic fatigue syndrome (ME/CFS).” Journal of translational medicine vol. 16,1 268. 1 Oct. 2018, doi:10.1186/s12967-018-1644-y Good luck.

I would recommend that the number of tables be reduced to three by consolidating the other two tables.  The figure is ok. 

Author Response

Thankyou for your review. 

The patients were selected for inclusion if they fulfilled the CDC criteria of Fukuda, 1994, and did not suffer from psychiatric illness, and did not smoke, etc. This is outlined in the microarray study paper (reference no. 5). In one of these, ME/CFS was triggered by laboratory documented acute EBV infection (VCA IgM positive and PCR positive for EBV DNA); this was one of the patients with a high level of EBI2. EBI2, a human gene, was found to be the most upregulated gene during EBV infection of Burkitt lymphoma cells (Birkenbach et al, 1993; reference no. 4), and upregulation of EBI2 is therefore accepted to indicate ongoing EBV replication/reactivation (references included in Reference no. 3). Therefore, case ascertainment did not depend on EBV markers. I have referenced our original paper (reference no. 5) for details of case ascertainment. 

The role of the EGR genes in EBV replication has been outlined in Table 3 (EGR genes in EBV infection), as well as on page 11 under section 6, entitled EGR genes in EBV infection. These contain specific reference to EGR genes and the EBV lytic cycle. 

In response to your suggestion, I have added a paragraph to the Conclusion to address the issue of whether the EGR genes could be used as biomarkers of either ME/CFS or EBV-associated ME/CFS. I don't think they would be so useful as biomarkers, but they seem to provide support for an ongoing role for EBV in the pathogenesis of some cases of ME/CFS. As ME/CFS is heterogeneous, I believe this is very interesting. 

I have added the reference you suggest (Rasa et al, 2018) as reference no. 163, with the citation in the (new) first line of the Conclusion. This is very relevant indeed. 

I would prefer not to combine 2 tables if I can avoid it, as some information would be lost as a consequence. I believe it is useful to view the information from the perspective of both EGR genes in EBV infection, and EGR gene functions in ME/CFS. However, if I have breached a limit, I will do my best to modify the manuscript.

Reviewer 3 Report

Despite the fact that Myalgic Encephalomyelitis/Chronic Fatigue Syndrome (ME/CFS) is a serious chronic, debilitating disease that affects 1 to 2.5 million Americans, the lack of the progress in ME/CFS research is still very notable. The problems can be attributed to a range of factors, including the paucity of large, high quality, hypothesis-driven studies, and controversy around diagnosis. However, the classification of CFS/ME as a chronic flu-like illness triggered by virus infection is sufficient enough. Several human herpesviruses including HHV-4 (Epstein-Barr virus, EBV) have been implicated in the pathogenesis of ME/CFS. EBV is a ubiquitous gammaherpesvirus that is highly prevalent in almost all human populations and is associated with numerous human disorders.

In the review article “Early Growth Response gene up-regulation in Epstein-Barr virus (EBV)-associated Myalgic Encephalomyelitis/Chronic Fatigue Syndrome (ME/CFS)” by Jonathan R Kerr, the author evaluates the up-regulation of EGR1, EGR2 and EGR3 in ME/CFS patients who exhibit upregulation of EBV-induced G-protein coupled receptor EBI2 as well. The hypothesis is that the up-regulation of these two sets of genes in ME/CFS is associated with the activity of the lytic cycle of EBV infection (reactivation).

Our data on a possible connection between herpesviruses such as EBV, and a heterogeneous autoimmune-like disease such as ME/CFS are still controversial. At the same time, considering that the patients suffering from ME/CFS are likely to be infected with EBV, - any new findings extend our understanding of such a complicated disorder. Therefore this analytical review will attract without doubt a broad general interest.

Author Response

Thankyou for your review. 

Yes, the hypothesis is that up-regulation of these genes (EBI2 and EGR genes) in ME/CFS is associated with the activity of the lytic cycle of EBV infection (reactivation). Upregulation of the EBI2 gene appears to be the biomarker of EBV-induced ME/CFS, which we have been looking for for some time. 

No modifications were suggested. 

Thankyou.